



# Technical note: Investigating sub-city gradients of air quality: lessons learned with low-cost PM2.5 and AOD monitors and machine learning

Michael Cheeseman[1], Bonne Ford[1], Zoey Rosen[2], Eric Wendt[3], Alex DesRosiers[1], Aaron J. Hill[1], Christian L'Orange[3], Casey Quinn[3], Marilee Long[2], Shantanu H. Jathar[3], John Volckens[3], Jeffrey R. Pierce[1]

[1]Department of Atmospheric Science, Colorado State University, Fort Collins, 80521, US
[2]Department of Journalism & Media Communication, Colorado State University, Fort Collins, 80521, US
[3]Department of Mechanical Engineering, Colorado State University, Fort Collins, 80521, US

*Correspondence to*: Michael Cheeseman (cheesemanmj@gmail.com)

**Abstract.** Accurate sub-city fine particulate matter (PM2.5) estimates could improve epidemiological and health-impact studies in cities with heterogeneous distributions of PM2.5, yet most cities globally lack the monitoring density necessary for sub-city-scale estimates. To estimate spatiotemporal variability in PM2.5, we use machine learning (Random Forests; RFs) and concurrent PM2.5 and AOD measurements from the Citizen Enabled Aerosol Measurements for Satellites (CEAMS) low-cost sensor network as well as PM2.5 measurements from the Environmental Protection Agency's (EPA) reference monitors during wintertime in Denver, CO, USA. The RFs predicted PM2.5 in a 5-fold cross validation (CV) with relatively high skill (95% confidence interval R2=0.74-0.84 for CEAMS; R2=0.68-0.75 for EPA) though the models were aided by the spatiotemporal autocorrelation of the PM2.5 measurements. We found that the most important predictors of PM2.5 were factors associated with pooling of pollution in wintertime, such as low planetary boundary layer heights (PBLH), stagnant wind conditions, and, to a lesser degree, elevation. In general, spatial predictors were less important than spatiotemporal predictors because temporal variability exceeded spatial variability in our dataset. Finally, although concurrent AOD was an important predictor in our RF model for hourly PM2.5, it did not improve model performance with high statistical significance. Regardless, we found that low-cost PM2.5 measurements incorporated into an RF model were useful in interpreting meteorological and geographic drivers of PM2.5 over wintertime Denver. We also explored how the RF model performance and interpretation changes based on different model configurations and data processing.

## 1 Introduction

Exposure to high concentrations of airborne particulate matter, especially particles with aerodynamic diameters smaller than 2.5 μm (PM2.5), has adverse effects on public health (Forouzanfar et al., 2016; Hennig et al., 2018; Lelieveld et al., 2019; Pope et al., 2002; Schwartz et al., 1996). Increased exposure to PM2.5 also imposes large economic burdens due to medical costs, welfare loss, disruptions to work productivity, and elevated crime rates (Burkhardt et al., 2019; Dechezleprêtre et al., 2019). As PM2.5 concentrations are generally higher in urban areas, this burden can be especially large in major cities (Anenberg et al., 2019; Marlier et al., 2016). Research has shown that urban concentrations of PM2.5 can be uniform with relatively larger



heterogeneity in black carbon, organic aerosol, and particle number concentrations (Gu et al., 2018; Saha et al., 2021). In addition, there can still be sub-city spatial and sub-daily temporal gradients in PM$_{2.5}$ that are difficult to measure due to the low spatial density of reference monitoring networks(Just et al., 2015; Bi et al., 2019; Gao et al., 2015; Wang and Oliver Gao,

2011). Improving predictions of PM$_{2.5}$ across cities could aid epidemiological investigations into the public health impacts of poor air quality (Southerland et al., 2021).

Low-cost sensor networks have been increasingly used to supplement reference networks and increase the spatiotemporal density of PM$_{2.5}$ measurements (Bi et al., 2020; Gao et al., 2015; Gupta et al., 2018; Snyder et al., 2013). These networks can

be deployed by citizen scientists, thus simultaneously contributing to our understanding of air pollution and increasing public awareness of air quality issues (Ford et al., 2019; Gupta et al., 2018). For example, the PurpleAir network (https://www.purpleair.com), which uses light-scattering sensors to estimate PM$_{2.5}$ at sub-hourly timescales, has thousands of citizen-deployed monitors across the US and has been growing rapidly over recent years (Delp and Singer, 2020; Krebs et al., 2021). Despite the usefulness of low-cost sensor networks, they are often limited by their lower quality monitors, which can

result in moderate to large uncertainties in their measurements (Gupta et al., 2018; Snyder et al., 2013). Furthermore, many regions in the US lack both low-cost and reference measurements of PM$_{2.5}$, which limits our understanding of public exposure to air pollutants.

Satellite observations of aerosol optical depth (AOD), an estimate of light extinction due to aerosols in an atmospheric column,

can provide near-global coverage of clear-sky regions every 1-2 days; these observations are useful for filling in the gaps of PM$_{2.5}$ monitoring networks. Since satellite-retrieved AOD does not provide information about surface PM$_{2.5}$ directly, various techniques have been developed to leverage AOD measurements to inform surface PM$_{2.5}$ predictions. These techniques can generally be grouped into two categories: geophysical and statistical approaches. The geophysical approach to translate satellite AOD into PM$_{2.5}$ uses chemical transport models (CTMs) to simulate the relationship between PM$_{2.5}$ and AOD (Hammer et al.,

2020; Liu et al., 2004, 2005; van Donkelaar et al., 2006, 2013, 2011) on global to local scales. The modeled PM$_{2.5}$:AOD ratios are then multiplied by the satellite AOD to derive an estimate of PM$_{2.5}$. While this approach is useful for annual-average concentrations (Hammer et al., 2020; van Donkelaar et al., 2010) and on shorter timescales for some locations and seasons (van Donkelaar et al., 2012), there are many limitations to this approach. For example, most satellites that capture AOD are polar-orbiting satellites, which only provide coverage during specific daytime-only (and cloud-free) overpass times, and hence

fully rely on the model's predicted diurnal cycles for daily mean PM$_{2.5}$ estimates. Modeled PM$_{2.5}$:AOD relationships have also been found to be a large source of uncertainty in satellite-derived PM$_{2.5}$ (Ford and Heald, 2016; Jin et al., 2019), and a lack of reference measurements of PM$_{2.5}$:AOD means they are difficult to validate. Monitoring networks such as the (SPARTAN) have been developed to provide high fidelity PM$_{2.5}$:AOD observations but the monitoring sites are expensive and there are few worldwide (Snider et al., 2015, 2016). Finally, the resolution of satellite AOD measurements and CTM grid cells tends to be

too coarse (e.g., >10 km) to study the fine-scale spatiotemporal resolutions necessary to capture the heterogeneity of PM$_{2.5}$


concentrations in urban areas, although recent satellite AOD products (Lyapustin et al., 2018) and high-resolution simulations (Jena et al., 2021; Kirwa et al., 2021) may remedy these issues.

Alternatively, satellite AOD retrievals can be incorporated into a statistical model to estimate surface $PM_{2.5}$. The simplest of these approaches uses co-located satellite AOD and surface $PM_{2.5}$ measurements in a linear regression model (Engel-Cox et al., 2004; Koelemeijer et al., 2006). However the relationship between AOD and $PM_{2.5}$ is complex and can vary due to changes in the aerosols' vertical distribution, water content, speciation, optical properties, and size distribution (Ford and Heald, 2016; Snider et al., 2015; van Donkelaar et al., 2010, 2006, 2013). Thus, many techniques for $PM_{2.5}$ estimation have been developed to incorporate information from many data sources including but not limited to AOD, meteorology, and geographic information such as land-use regression (Hoogh et al., 2016; Song et al., 2014) and geographically weighted regression (e.g. Lassman et al., 2017), not all of which are inherently suited to estimate both spatial and temporal variability in $PM_{2.5}$ concentrations. Even more complex computational and machine learning (ML) methods are also becoming increasingly common in estimating $PM_{2.5}$ (e.g. Di et al., 2016; Lightstone et al., 2017; Liu et al., 2018; Reid et al., 2015; Suleiman et al., 2019; Xi et al., 2015). Already, ML methods have been shown to be more accurate at predicting $PM_{2.5}$ than traditional CTM methods under certain conditions (Lightstone et al., 2017; Xi et al., 2015), and they can require less expertise to operationalize than CTMs.

ML represents a range of computational methods that build predictive models without explicit programming and with limited human intervention. One of the benefits of ML methods is that most can capture complex, non-linear relationships between many predictors (e.g., wind speed, AOD, land use) and a target variable (in this case, $PM_{2.5}$) in order to produce explicit predictions of the target variable. Generally, ML models find relationships between predictors and the target using a training dataset, which is then validated using an independent testing dataset. Although the training and testing process is done with little human interference, the complexity and flexibility of ML models must be decided beforehand, and it is difficult to know what model configurations will result in the highest prediction skill. Thus, models must be tuned to find optimal configurations that reduce the risk of overfitting or underfitting the training dataset. As ML methods become more widely used, transparency in the execution of these methods and how they are validated will be key for the research community to ensure the quality of results obtained.

In this work, we use ML methods to investigate spatiotemporal variability in wintertime Denver. This work uses low-cost sensor measurements from the Citizen Enabled Aerosol Measurements for Satellites (CEAMS) project in addition to regulatory $PM_{2.5}$ measurements. The CEAMS project has (1) developed a low-cost monitor that can capture sub-hourly coincident $PM_{2.5}$ and AOD measurements and (2) trained citizen scientists to deploy them to study fine-scale spatiotemporal variability in the relationship between $PM_{2.5}$ and AOD. The CEAMS team conducted a deployment of these monitors during the winter of 2019-2020 in Denver, Colorado, United States (hereafter just "Denver"). To our knowledge, this was the first high-density network of low-cost, coincident sub-hourly AOD and $PM_{2.5}$ sensors deployed in a single city. We investigate the potential drivers of





fine-scale PM$_{2.5}$ spatiotemporal variability in wintertime Denver by incorporating meteorological and geographical variables into a random forest (RF) ML regression framework (Breiman, 2001). We use a permutation metric to assess the relative importance of different predictor variables. We test whether co-located AOD measurements are identified as an important predictor of PM$_{2.5}$ and whether they increase the overall RF prediction skill compared to RFs that only used geographic and meteorological variables. The RF method was used here because it has been used to skillfully estimate PM$_{2.5}$ in past studies

(Reid et al., 2015; Considine et al., 2021). We also compare our analysis of CEAMS PM$_{2.5}$ with results using reference PM$_{2.5}$ measurements from the Environmental Protection Agency (EPA). Finally, we discuss our RF methods in detail and discuss how decisions made during data processing and model configuration may have influenced our results and the subsequent interpretation.

## 2 Methods

### 2.1 Data Sources

### 2.1.1 CEAMS PM$_{2.5}$ and AOD dataset

The CEAMS team developed two generations of low-cost monitors called the Aerosol Mass and Optical Depth (AMOD) monitors (Wendt et al., 2021). The AMODs used in this study are second-generation instruments (i.e., AMOD-v2) but, as the first version is no longer in use, we simply refer to the devices as AMODs herein. The CEAMS team trained citizen scientists

to deploy AMODs in several different campaigns in northern Colorado (e.g. Ford et al., 2019). Here we analyze data from the CEAMS deployment during the winter of 2019-2020 in Denver (Fig. 1). Thirty-two participants were recruited from across Denver through collaboration with the Community Collaborative Rain, Hail and Snow (CoCoRaHS) citizen scientist network (Cifelli et al., 2005) and other media outreach. Participants were trained by CEAMS researchers to set up devices using a mobile application (Quinn et al., 2019) and replace aerosol filters once a week. Measurements were taken from 14 November

2019 to 20 January 2020.





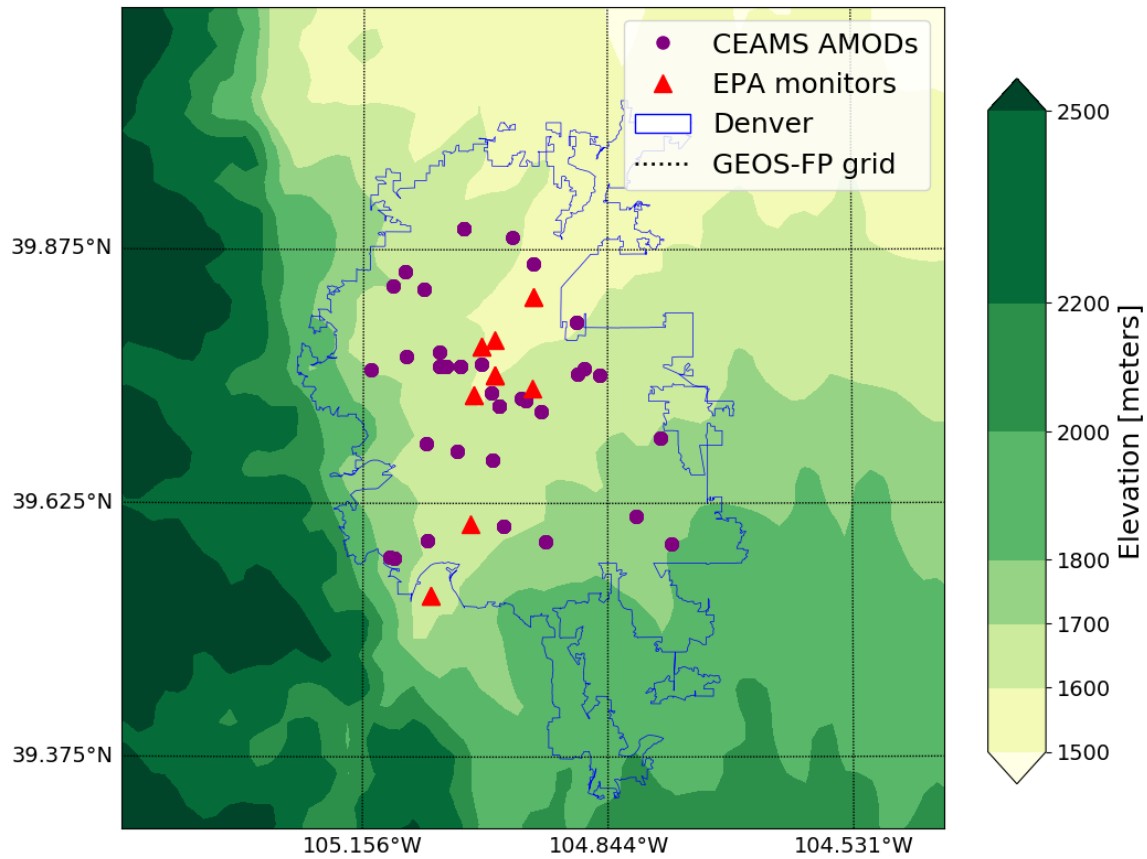

**Figure 1: Map of elevation (Amante and Eakins, 2009) over Denver, CO, with locations of CEAMS Aerosol and Mass Optical Depth (AMOD) monitors (purple), EPA reference PM$_{2.5}$ monitors (red), outlines of the GEOS-FP grid-boxes (black), and outlines of the greater Denver-Aurora area (blue; based on cartographic files from 2015 TIGER/Line Shapefiles).**

The AMOD is a low-cost (~ $1,200 manufacturing cost) PM$_{2.5}$ and AOD monitor that measures PM$_{2.5}$ in two ways: (1) real-time measurements using an inexpensive light-scattering sensor and (2) time-integrated measurements by collecting particles onto a filter using a size-selective cyclone separator and an ultrasonic pumping system (Volckens et al., 2017; Wendt et al., 2019, 2021). The real-time PM$_{2.5}$ sensor is the Plantower PMS5003, which has been widely deployed by networks such as PurpleAir, and validated in past work (Bulot et al., 2019; Sayahi et al., 2019). The AMOD also measures AOD at four discrete

wavelengths (440, 500, 675, and 870 nm) using optically filtered photodiodes. The AMOD uses a solar tracking procedure that allows for automated AOD measurements throughout the day (Wendt et al., 2021). When AMODs were co-located with Aerosol Robotics Network (AERONET) AOD monitors in a series of validation experiments, the mean absolute error was 0.057 over AOD values ranging from 0.030 to 1.51 (Wendt et al., 2021). Real-time PM$_{2.5}$ and AOD can be sent to a central server by the AMOD over Wi-Fi every 20-minutes. The real-time PM$_{2.5}$ values used in this study were an average of

instantaneous 1s values reported every 15-seconds over a period of 2.5 minutes (after a 30 second warm up period), taken every 20 minutes.





The real-time $PM_{2.5}$ data were quality controlled for possible sources of error and bias. First, any real-time $PM_{2.5}$ measurement reported over 500 $\mu$g m$^{-3}$ was removed based on the manufacturer's guidance, similar to (Lu et al., 2021). Second, the AMOD

$PM_{2.5}$ measurements were aggregated in two different temporal resolutions: 1-hour and 24-hour averages. The 24-hour averaged $PM_{2.5}$ measurements were only used if there were measurements for at least ¾ of the day to ensure it was representative of the entire day. Third, to correct for known biases of Plantower data tied to relative humidity (RH), we applied the following simple additive model in Eq. (1) tested by the US EPA (Barkjohn and Clements, 2020):

$$PM_{2.5} = 0.524 \times AMOD_{CF1} - 0.0862 \times RH + 575 \tag{1}$$

The Plantower reports multiple $PM_{2.5}$ values based on different corrections, but $AMOD_{CF1}$ refers to $PM_{2.5}$ reported by the Plantower that was not corrected by the manufacturer's built-in atmospheric correction. The RH values used to correct each $PM_{2.5}$ data point were taken from the Plantower sensor as well. The Barkjohn and Clements (2020) model was developed specifically for the low-cost PurpleAir PM monitoring network, which uses either the PMS5003 or PMS7001 Plantower sensors. In this study, the Plantower $PM_{2.5}$ data were not corrected using the time-integrated filter measurements of $PM_{2.5}$ taken

by the AMODs as in Ford et al., (2019).

The AMOD 500 nm AOD data were quality controlled based on a procedure previously described by Ford et al., (2019) that is based on similar methods used by AERONET. As long as the sun is greater than 10 degrees above the horizon (estimated by the solar tracking algorithm), the device will attempt to take 3 AOD measurements, or a triplet, within a 1-minute period at

the start of each 20-minute interval. We did not require that AOD was measured over ¾ of the day, as we did with $PM_{2.5}$, since successful AOD measurements were less frequent and similar measurements from satellites only capture 1-2 times per day. Quality control and cloud screening were then applied in post-processing on each triplet at each wavelength. If less than 2 measurements per triplet attempt were taken or the range of AOD values was too large (>0.02) at any wavelength, then no measurements from that interval were used in this analysis. AOD was also filtered to remove measurements with air mass

factors > 5 or an Ångström exponent < 0. The Angstrom exponent was measured between the 440 nm and 875 nm wavelengths. Finally, we assumed that 500 nm AOD values that were outside of the range 0-1 were likely the result of measurement errors, such as cloud contamination, though we acknowledge this may be wrong for dust or smoke-impacted scenes (which are uncommon in wintertime Denver).

### 2.1.2 EPA $PM_{2.5}$ measurements

We used 24-hour averaged $PM_{2.5}$ measurements from the Environmental Protection Agency's (EPA) Air Quality System (AQS) network from eight sites based in Denver (https://aqs.epa.gov/aqsweb/airdata.html) as shown in Figure 1. We limited our analysis to the eight sites that use federal reference methods or federal equivalent methods for $PM_{2.5}$ and report local conditions (EPA parameter code 88101). We show in Table S1 the characteristics of each EPA monitoring site used in this



study. Since EPA AQS $PM_{2.5}$ data is available for multiple years, we analyze data from November 1st - January 31st for the
winters of 2017-2018, 2018-2019, 2019-2020. In Section 3, we choose to show results using the three years of data rather than
limiting to the time period of the CEAMS deployment. However, we did the analysis for both time periods, and as we will
discuss, the EPA RF results are similar, though noisier, if we limit EPA data to the time period of the CEAMS deployment.

### 2.1.3. Spatiotemporal ML predictor datasets

We used meteorological data (Table 1) from the Goddard Earth Observing System forward-processing dataset (GEOS-FP)
provided by the Global Modeling and Assimilation Office. GEOS-FP is produced with a native resolution of 0.25° (longitude)
x 0.3125° (latitude) (~25 km horizontal resolution) with 72 hybrid vertical layers . The GEOS-FP data used were hourly or 3-
hourly time averaged, depending on the variable. The 3-hourly data were linearly interpolated to hourly time resolution.
Finally, all of the GEOS-FP variables were linearly interpolated spatially to the CEAMS and EPA monitor locations to better
relate $PM_{2.5}$ observations with the environment for RF predictions.

### 2.1.4. Spatial-only ML predictor datasets

We used multiple spatially varying datasets to describe each CEAMS and EPA monitoring location. Elevation information at
each monitor location was extracted from the Global Multi-resolution Terrain Elevation Data 2010 (GMTED2010) provided
by the U.S. Geological Survey (USGS) and the National Geospatial-Intelligence Agency (NGA) (Danielson and Gesch, 2011).
The GMTED2010 data used in this analysis were at a 15-arc second (450 meters) horizontal resolution, which provides a
unique elevation value for each CEAMS AMOD and EPA site. The slope of the terrain at the dataset resolution was calculated
from GMTED's elevation data using QGIS. Additionally, we considered two predictors that have been found to be significant
in land use regression (LUR) modeling: population density and travelled miles. We used census tract population density
estimates from the Colorado Department of Public Health and Environment (CDPHE) (https://data-
cdphe.opendata.arcgis.com/) as a predictor in our RF models. Population density estimates on the CDPHE site, measured in
population density per square mile of land area within the given census tract, are directly from the 2013-2017 American
Community Survey. Finally, we also incorporated traffic and road density data into our RF $PM_{2.5}$ predictions. We used model-
assigned 2020 All-day Traffic Volumes from the Denver travel model, "Focus" (Model Cycle: RTP-2020 and Focus 2.3;
https://drcog.org/services-and-resources/data-maps-and-modeling/travel-modeling) developed by the Denver Regional
Council of Governments. This data includes shapefiles of large, medium, and small (i.e., arterial, collector, and local) road
segments and a model estimate of annual average traffic volumes on each segment, which is measured in vehicles that traveled
on each segment per year. We determined the length of each road segment within a 500 buffer (i.e., intersection length) around
each CEAMS and EPA monitor and then multiplied the intersection lengths by the traffic volumes of each segment, thereby
producing an estimate of miles traveled by vehicles per year within 500 m of each monitor. As will be shown later, both
population density and traveled miles were not found to be significant in estimating variability in $PM_{2.5}$ and hence we decided



not to include any more LUR predictors in our model. A more comprehensive dataset of LUR predictors could be used in conjunction with geographical and meteorological predictors in future RF modeling.

## 2.2. Random Forest Models

### 2.2.1.  Random Forest set-up and predictors

To investigate the complex relationships between meteorology, geography, and air quality as well as the value that AOD can
add to predicting air quality, we used RF ML regression models. RF models are made up of a group of unique and weakly correlated decision trees that are leveraged together to make a prediction. A decision tree begins with a random subset of the training data at the first node (i.e., the root node) and successively splits the data into branch nodes (Figure 2). Here, a mean value of $PM_{2.5}$ is predicted to represent all the samples in each node and the mean squared error is found. The data samples are then split at each branch node using a true or false question about one of the predictors (e.g., "is the temperature > 290 K?"),
which is chosen to reduce the mean squared error of the samples in the following nodes. This process continues until the decision tree reaches its termination criterion, such as when there are not enough samples to form a new branch node or the tree depth (i.e., number of branches) reaches some maximum set beforehand. At this point leaf nodes are formed with final predictions. These trees are built during the training process and then the testing data will follow the split nodes until they arrive at leaf nodes, which provide predictions for each value. This process is repeated using each tree in the forest, and the
final prediction for each testing sample is given as an average of the predictions across all of the trees. The strength of RF models is that they leverage predictions from many weakly correlated decision trees, which helps protect the model against biases. The RF ensures that decision trees are weakly correlated and unique by giving a random subset of the samples and predictors to each decision tree, and another random subset of the predictors to choose from at each branch node during the training process. In this study, we created RF models with the scikit-learn Python package (Pedregosa et al., 2011) to predict
the spatial and temporal variability of $PM_{2.5}$ using the predictors in Table 1.



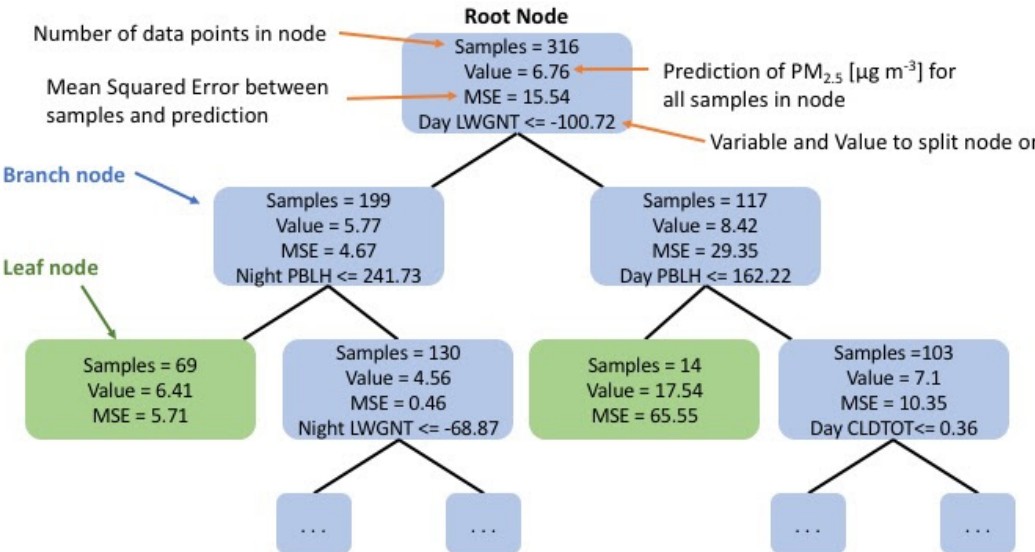

**Figure 2. Example schematic of a random forest decision tree. The root and branch nodes are in blue, while the leaf nodes, which hold the final predictions, are in green. The black lines represent a splitting of a branch node into two more nodes.**

In addition to investigating the relative importance of all of the different variables to predicting wintertime Denver PM$_{2.5}$, we

also wanted to specifically test the skill that AOD adds to the prediction of PM$_{2.5}$. Thus, we created 3 subsets of both hourly and 24-hour CEAMS data, resulting in 6 datasets. The first subset, called the "Full Dataset," used all hourly or 24-hour CEAMS PM$_{2.5}$ data points and the co-located meteorological and geographic variables; however, it did not use the CEAMS AOD as a predictor. The second subset, called "Test - AOD," only used hourly or 24-hour CEAMS PM$_{2.5}$ at times and sites where CEAMS AOD was also available. However, similar to the first subset, the second subset did not use AOD as a predictor.

Finally, the third subset, called "Test + AOD," is the same as the second, but CEAMS AOD was used as an additional predictor. Using these 3 subsets allowed for the investigation of three questions: 1) What is the change in prediction skill of our models if we limit the data to locations and days where AOD is available but we do not use AOD as a predictor? 2) When we use AOD as an additional predictor, how important is it for predicting PM$_{2.5}$ over wintertime Denver using the permutation metric? 3) How does the overall RF model skill change for predicting PM$_{2.5}$ after AOD is included as a predictor? Using models with

both hourly and 24-hour data allowed us to analyze the relationships among air quality, meteorological and geographical factors, and the prediction skill of AOD measurements at different timescales.

To compare the CEAMS RF results to reference measurements, an additional RF model was created to predict 24-hour averaged EPA PM$_{2.5}$. We used the same predictors in our EPA RF model as we did with the CEAMS data, except for AOD as

the EPA monitors do not have co-located AOD monitors. While there are fewer EPA monitors, they provide three full winters of data, allowing us to test whether our conclusions are robust when applied to a longer time period.



**Table 1. Predictor variables used in our RF models. A check-mark indicates when a predictor was used in a given model. The number of data points used for the training and testing of each RF model is listed at the bottom. A dash is used to indicate that AOD was not used as a predictor or that a column is not applicable.**

| Predictor | Description | Units | Data Source | CEAMS Full dataset | CEAMS Test - AOD | CEAMS Test + AOD | EPA model |
|---|---|---|---|---|---|---|---|
| Temp[†] | Surface skin temperature | Kelvin | GEOS-FP | ✓ | ✓ | ✓ | ✓ |
| RH[†] | Relative humidity | % | GEOS-FP | ✓ | ✓ | ✓ | ✓ |
| Wind Speed[†] | Wind speed at 10 meters height | m s$^{-1}$ | GEOS-FP | ✓ | ✓ | ✓ | ✓ |
| U$^{*}$[†] | Friction velocity | m s$^{-1}$ | GEOS-FP | ✓ | ✓ | ✓ | ✓ |
| Precip[†] | Precipitation total | inches | GEOS-FP | ✓ | ✓ | ✓ | ✓ |
| Cloud Frac[†] | Cloud total fraction | N/A | GEOS-FP | ✓ | ✓ | ✓ | ✓ |
| LWGNT[†] | Longwave net radiation | W m$^{-2}$ | GEOS-FP | ✓ | ✓ | ✓ | ✓ |
| SWGDN[†] | Shortwave downwelling radiation | W m$^{-2}$ | GEOS-FP | ✓ | ✓ | ✓ | ✓ |
| PBLH[†] | Planetary boundary layer height | meters | GEOS-FP | ✓ | ✓ | ✓ | ✓ |
| Elevation | Elevation above sea level | meters | GMTED 1 km grid | ✓ | ✓ | ✓ | ✓ |
| Slope | Slope of terrain | degrees | calculated from GMTED | ✓ | ✓ | ✓ | ✓ |
| Pop density | Population density | People per square miles | US Census | ✓ | ✓ | ✓ | ✓ |
| Traveled miles | Annual average miles traveled within 500 meters of monitor | miles year$^{-1}$ | Focus 2.3 model | ✓ | ✓ | ✓ | ✓ |
| CEAMS AOD | Daily mean 500nm AOD | dimensionless | CEAMS AMODs | - | - | ✓ | - |





| Number of data points in 24-hour models: | 634 | 307 | 307 | 2411 |
|---|---|---|---|---|
| Number of data points in hourly models: | 18969 | 1043 | 1043 | - |

† Daytime (11am-3pm) and nighttime (11pm-3am) averages of these variables were used as separate predictors when using the 24-hour averaged PM$_{2.5}$ from the CEAMS and EPA dataset. For hourly data, these predictors were taken from the same location and hour of the PM$_{2.5}$ data they are matched with. One exception is that nighttime-averaged SWGDN was not used as there is no shortwave downwelling radiation at night.

### 2.2.2. Model configuration and tuning

When implementing a ML technique, such as RFs, models must be appropriately tuned. Tuning is the process for configuring the structure and assumptions of the model. For RF models specifically, the tuning process generally controls the number and complexity of the decision trees that make up the random forest, the way in which data are sampled, and the number of predictors that should be considered at each split in each tree (i.e., hyperparameters). Tuning is necessary to ensure that the model does not underfit or overfit the training data. For an RF model, overfitting the training data occurs when the decision

trees begin fitting onto the noise that exists in the training dataset. As a consequence, the RF could skillfully re-create the PM$_{2.5}$ values of the training dataset if fed the same predictor values associated with those training values (i.e., the same combination of meteorology, geography, etc.) because it learned to predict even the noise of the training data. However, if this same RF model was given an unseen set of PM$_{2.5}$ values and their associated predictors, such as PM$_{2.5}$ from a different time period or a different monitor, it could perform poorly. Overfitting can go unnoticed if there is data leakage between the testing and training

data, which could occur if the data in both sets are autocorrelated. Underfitting, on the other hand, is more straightforward; it occurs when the decision trees are too simple and fail to capture the relationships that exist between the predictor and target variables. Typically, to ensure against over or underfitting, the data are split into separate tuning, training, and testing datasets. However, since our dataset spans only a couple months, we tuned, trained, and tested our RF models using a cross validation (CV) method.


As ML methods become popular in air-quality research, we hope that transparency about our tuning process allows for reproducibility and serves as a guide for future work. We used a *k*-fold cross validation method to tune each model over a selection of hyperparameters (Table 2) using the scikit-learn package GridSearchCV (Pedregosa et al., 2011). GridSearchCV automatically trains and validates an RF model for every combination of hyperparameters given to it (Table 2) over *k* number

of folds (in our case, 5 folds for each hyperparameter combination). A *k*-fold CV entails chunking the data into *k* number of equally sized groups, using k-1 number of folds for training the RF model, validating that RF model using the remaining fold, and then repeating that process until every fold has been used for validation. We chose the final hyperparameters for our 6 CEAMS RF models and our EPA RF model based on the best MSE for each combination of hyperparameters in each model. However, if a similar MSE was found for a hyperparameter selection that allowed for simpler tree structures (shallower depth,

fewer trees, etc.), the simpler model was chosen instead of the more complex model. For example, Figure S1a shows that RF model skill when using 120 trees or 90 trees result in very similar distributions of model skill. Thus, while tuning the RF model



that used the 24-hour CEAMS Full Dataset, we chose 90 trees to limit the complexity of the model without losing performance. Similarly, we also chose a maximum depth of 15, 2 samples needed to form a leaf, and 5 samples needed to split a branch node. More information about the hyperparameters chosen for each RF model is in the Supplement (Fig. S1-S7).


**Table 2.  Hyperparameters tested during the tuning of each random forest model.**

| Hyperparameter in scikit-learn | Description | Values Tested |
|---|---|---|
| n_estimators | Number of trees | 20, 30, 40, 50, 60, 70, 90, 120 |
| max_depth | Maximum depth of each tree | Varies with different models |
| min_samples_split | Minimum samples needed to split an internal node | 1, 2, 5 |
| min_samples_leaf | Minimum samples needed to split at a leaf node | 1, 2, 4 |
| max_features | Maximum number of predictors to consider for each split | 'Sqrt', 'Auto' |
| bootstrap | Each decision tree will be built with a bootstrapped sample of the dataset | True, False |

### 2.2.3. Validation and bootstrapping methodology

Once the final model hyperparameters were chosen, the models were trained and tested over another 5-fold CV. Although the CV method was used in both our tuning and testing methodology, each was done using a different random shuffling of the
data. See Figure S8 for an example of a comparison between CEAMS 24-hour $PM_{2.5}$ and the RF prediction after validating against 1 testing fold. We estimated the uncertainty in our RF model predictions by calculating 95% confidence intervals for the performance metrics of each RF model using a bootstrapping method. Bootstrapping entailed taking random samples of the model predictions and the associated $PM_{2.5}$ measurements, with replacement, and finding the errors statistics (e.g. the root mean squared error [RMSE] and the coefficient of determination [$R^2$]) of each random sample. This process was repeated until
a distribution of each error statistic was created. Then the error statistics were sorted into ascending order and the values at the 2.5% and 97.5% percentiles represented the 95% confidence interval. Finally, to investigate the relative importance of each predictor for the RF predictions, a permutation importance metric was used, which tests the change in model prediction skill after randomly shuffling one predictor of the validation data at a time. Thus, the higher the permutation importance, the greater loss of prediction skill if that predictor was randomized.  To test the robustness of each permutation importance score, the
metric was calculated 100 times for each predictor for each of the 5 iterations of the 5-fold CV, resulting in a distribution of 500 permutation importance scores per predictor.



### 2.2.4. The impact of autocorrelation in RF methods

The CEAMS and EPA $PM_{2.5}$ measurements were autocorrelated at both hourly and 24-hour timescales. This lack of independence can result in information being shared between the training and testing datasets. This information sharing makes it much easier for the RF models to predict $PM_{2.5}$ from the testing dataset because it looks very similar to the data the models were trained on. Thus, the RF models' prediction skills can be inflated. In our models, this information sharing occurred when the $PM_{2.5}$ and predictor variables were randomly shuffled before we chunked the data into $k$-folds (i.e., "shuffled $k$-folds"), which is the default behavior of many $k$-fold CV procedures because they assume each value is an independent sample. Alternatively, data can be chunked into consecutive $k$-folds, which reduces information sharing between testing and training datasets (Fig. 3).

To understand the impact of autocorrelation on our models, we present results from training and validation with shuffled and consecutive $k$-folds in the Results and SI, respectively. Our analysis of the CEAMS data, which was limited to several weeks of measurements, showed larger differences between results using shuffled data and results using consecutive k-folds compared to our EPA analysis, as discussed in Section 3.4. This is likely due to the greater noise, residual RH bias of the Plantower sensors, and inconsistent sampling pattern of the citizen science deployment (Fig. S9), which limited the predictability of CEAMS $PM_{2.5}$. Thus, the CEAMS RF models using consecutive folds often performed poorly and our confidence in the chosen predictors was low. Hence, while the consecutive method is preferable for long, comprehensive data sets, we present here the CEAMS results of our RF models using shuffled data because we found that, even though their predictive ability appears inflated for unseen data due to the autocorrelation, they still allow for useful interpretations of meteorological, geographical, and other predictors of $PM_{2.5}$.

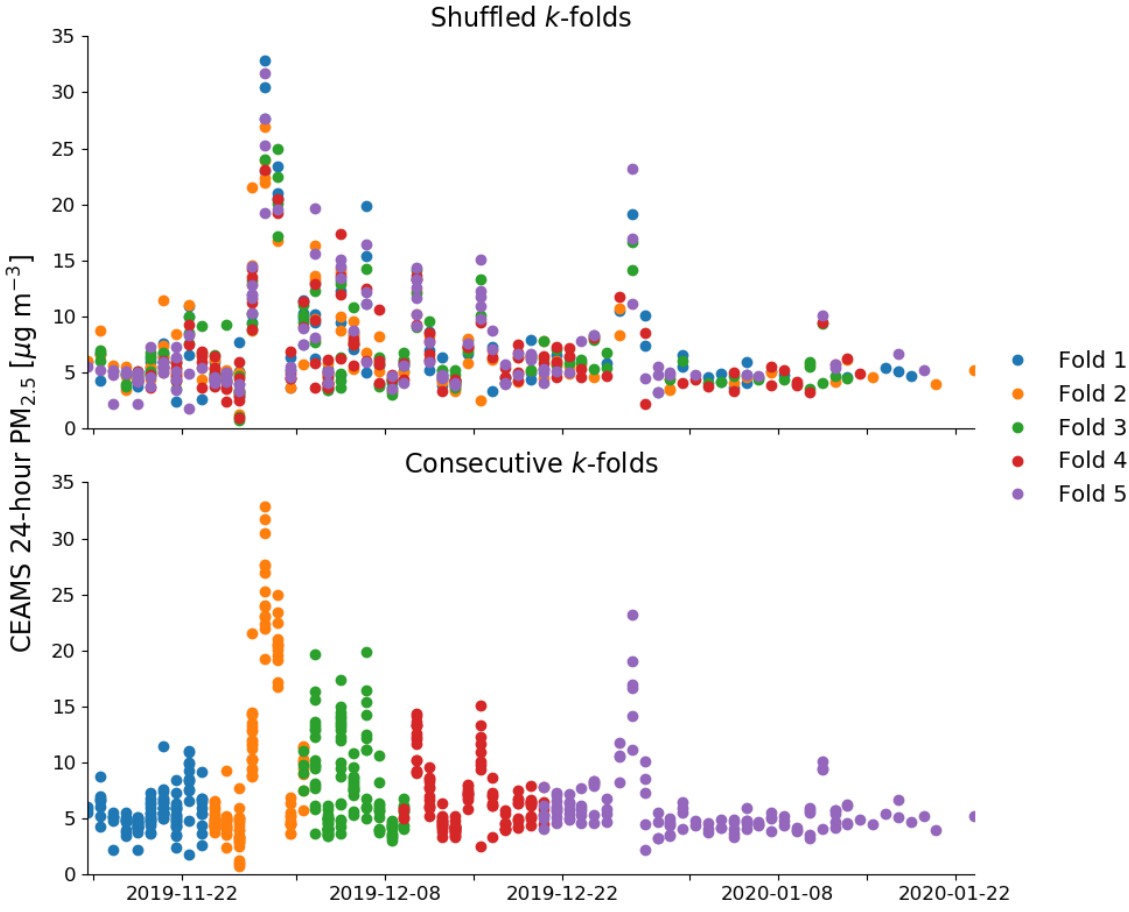

**Figure 3. Example of CEAMS 24-hour data split into 5 randomly shuffled folds (a) and 5 consecutive (i.e., chunked in time) folds (b) used in the RF model training and testing process. During a 5-fold CV, 4 of these folds are used to train a model and the remaining fold is used to validate the model, which is repeated iteratively until each fold has been used for validation.**

## 3. Results

### 3.1. CEAMS Denver Deployment Data

During the CEAMS pilot deployment in wintertime Denver our AMODs retrieved over 18,000 hourly averaged quality-controlled PM$_{2.5}$ measurements ($\mu$ = 8.2 µg m$^{-3}$; $\sigma$ = 12.6 µg m$^{-3}$) and over 1000 hourly averaged quality-controlled AOD measurements ($\mu$ = 0.06; $\sigma$ = 0.05) (Table 1). There were only a few periods of significantly elevated PM$_{2.5}$ (24-hour means > 10 µg m$^{-3}$) during the deployment (Fig. 4a), and they did not often coincide with a proportional increase in daytime AOD (Fig. 4b). Thus, there was a low correlation between PM$_{2.5}$ and AOD (Fig. 4c). The days with elevated 24-hour averaged PM$_{2.5}$ tended to be driven by late afternoon and overnight buildup of air pollution potentially caused by automobile emissions and



residential heating during stable winter nighttime conditions over Denver (Fig. S10). There were also strong sub-city gradients

of concentrations during some periods as shown in Figure S11.

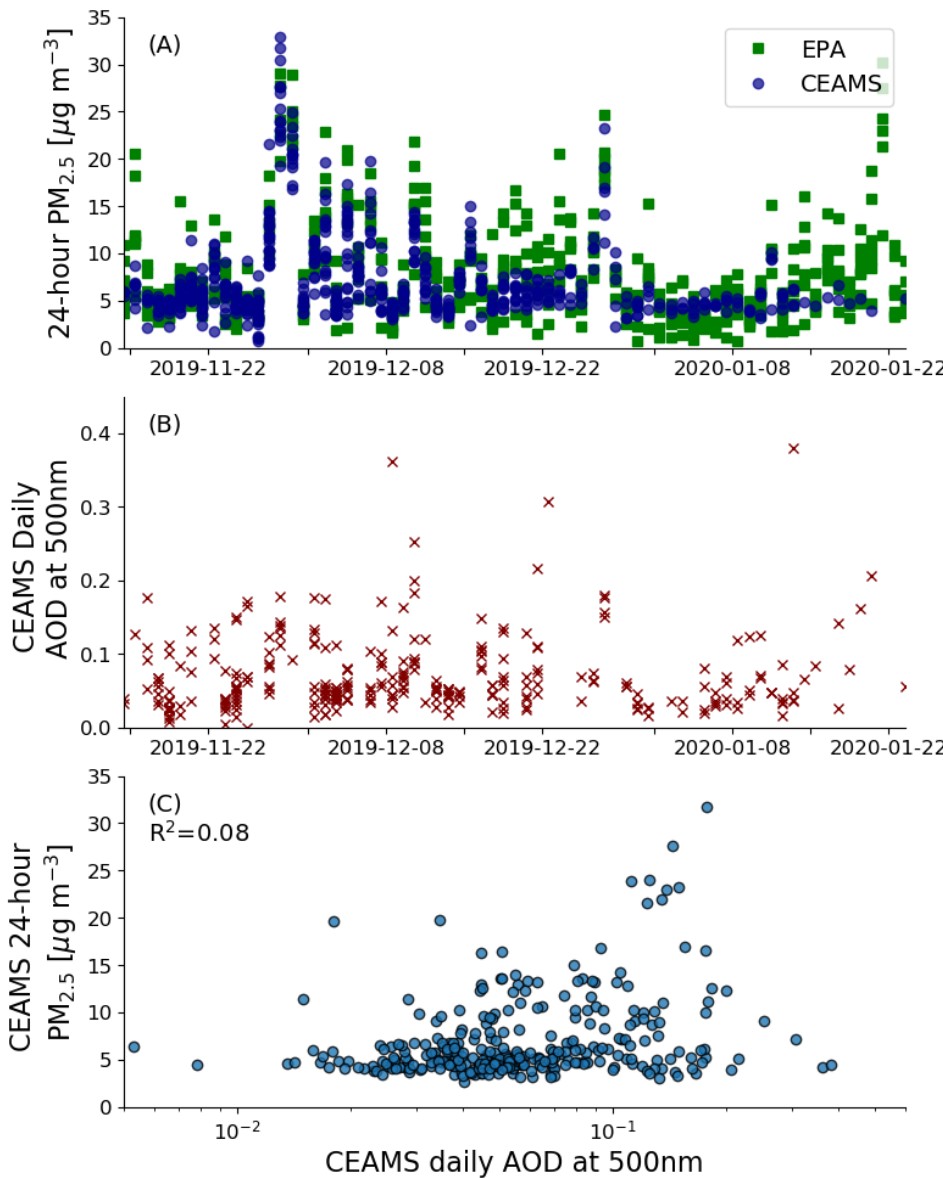

**Figure 4. (a) CEAMS AMODs' 24-hour averaged PM$_{2.5}$ measurements taken in Denver, CO by 32 citizen scientists between November 14th, 2019, and January 20th, 2020. EPA reference measurements of 24-hour PM$_{2.5}$ from 8 sites are also shown for the same period. (b) CEAMS AMODs daily averaged AOD measurements taken by the same devices shown in panel a. (c) The**
**relationship between 24-hour time averaged PM$_{2.5}$ and daily averaged AOD taken by the same CEAMS AMODs.**

To enhance our understanding of the potential drivers of PM$_{2.5}$ over wintertime Denver, prior to creating ML models, we

investigated the relationship between our CEAMS 24-hour PM$_{2.5}$ measurements and different spatial and spatiotemporal

predictors (Fig. 5). This analysis helps set expectations for potentially important predictors in the ML models. We found that





24-hour $PM_{2.5}$ was negatively correlated with daytime and nighttime planetary boundary layer heights (PBLH), friction

velocity (U*), wind speed, and nighttime temperature (all of which are positively correlated with each other). The correlation

between our $PM_{2.5}$ measurements and these meteorological predictors is likely due to wintry conditions in Denver that lead to

stagnant air, thermal inversions, and low boundary layers, which can all serve to slow the ventilation and downwind transport

of urban air pollution. We also hypothesize that wintry conditions also may have led to increased wood burning for residential

heat, which would enhance $PM_{2.5}$ build up, especially overnight. However, this temperature-emission connection is a

hypothesis that we do not test here. $PM_{2.5}$ tended to be elevated during higher RH conditions as well, which may be due to a

combination of the physical connection between $PM_{2.5}$ and meteorological conditions, as well as remaining RH bias in the

measurements that was not removed using the Barkjohn and Clements (2020) correction. We explore this RH connection more

in our discussion of variable importance in our CEAMS and EPA RF models. The spatial-only predictors (elevation, slope,

population density, and vehicle travelled miles) were only weakly correlated with $PM_{2.5}$ because temporal variability

dominated over spatial variability in our dataset; however, these spatial predictors may still provide information to refine the

ML estimates. Figure 5 shows that $PM_{2.5}$ likely has a complicated and nonlinear relationship with local meteorology during

our deployment. However, it is difficult to interpret which variables or combinations of variables are more useful for predicting

$PM_{2.5}$ which is why we chose to use RF models to quantitatively determine predictors of spatiotemporal variability of $PM_{2.5}$

in wintertime Denver.



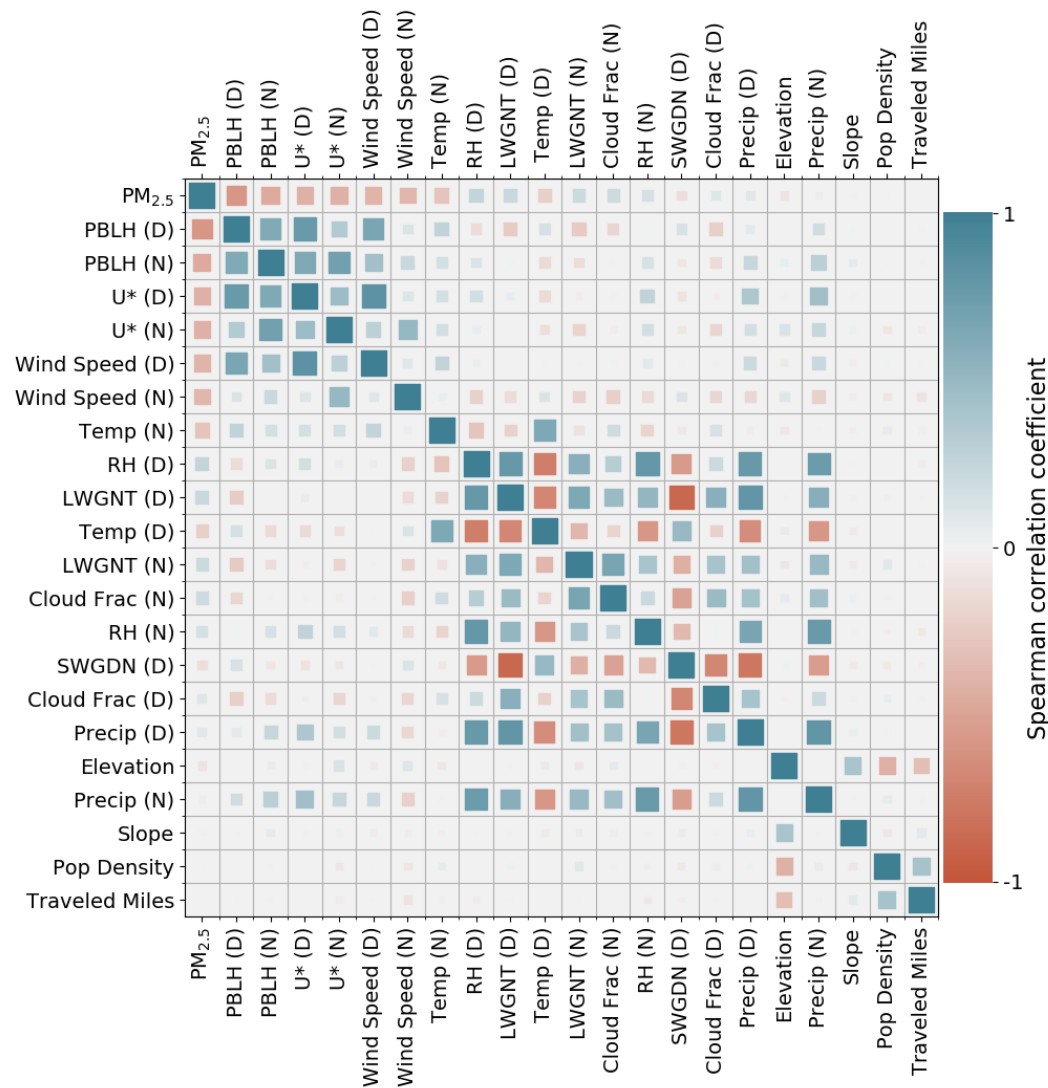


**Figure 5. Correlation matrix of CEAMS 24-hour PM₂.₅ data and all RF model predictors used in each 24-hour model. The "(D)" and "(N)" represent daytime (11am-3pm) and nighttime (11pm-3am) averages, respectively, of each meteorological predictor. The predictors are in order from greatest to least corresponding to the absolute value of the spearman rank correlation. The size of each box also corresponds to the absolute value of the spearman rank correlation between each variable so that the least important predictors have the smallest boxes. The same predictors are used in the RF models that predict hourly PM₂.₅ but hourly averaged meteorological factors were used instead of daytime and nighttime averages.**

### 3.2. Random Forest model skill

In Figure 6, we present the 95% confidence intervals of the performance metrics for each RF model using shuffled *k*-folds.

We found that, of the RF models predicting 24-hour PM₂.₅ measurements, the model using the CEAMS Full Dataset showed

the highest coefficient of determination, lowest RMSE, and a slope nearest to 1 between predictions and PM₂.₅ measurements.



The 95% confidence intervals of the Full Dataset model overlapped with the Test - AOD and Test + AOD RF models, which implies that limiting the CEAMS data to locations and days where AOD was available did not result in a significant reduction in model skill for 24-hour $PM_{2.5}$ prediction. We did see a small reduction in skill for hourly $PM_{2.5}$ predictions (Fig. 6e-g) after limiting the dataset to only locations and hours where AOD was also taken. The 24-hour CEAMS Full Dataset model also

showed similar skill to the EPA model for all metrics (Fig. 6a-d). However, results for the RF models using consecutive $k$-folds showed a significant decrease in prediction skill, especially for the CEAMS Full Dataset, while the EPA model results showed a less substantial decrease in skill (Fig. S12). We also found that the RF models were better at capturing temporal variability than spatial variability during the CEAMS deployment. The hourly $PM_{2.5}$ observations showed an average spatial standard deviation of ~2.5 $\mu$g m$^{-3}$ while the RF model predictions showed an average spatial standard deviation of ~1.5 $\mu$g m$^{-3}$

$^3$ for shuffled $k$-folds (Fig. S13) and only ~0.6 for consecutive $k$-folds (Fig. S14) .

By comparing the CEAMS Test - AOD and the CEAMS Test + AOD model performance metrics, we investigated the change in model performance if AOD was used as an additional predictor of $PM_{2.5}$. We found that the confidence intervals of the Test - AOD and Test + AOD models almost entirely overlapped for 24-hour $PM_{2.5}$ predictions (Fig. 6a-d), which shows that the

daily averaged AOD did not add to the overall prediction skill of the RF models. We found a small increase in mean model skill when comparing hourly $PM_{2.5}$ predictions between the Test - AOD and Test + AOD, indicated by the increased $R^2$, decreased RMSE, and a slope nearer to 1 for the Test + AOD model, but the confidence intervals overlap, which indicates that the difference between models had low statistical significance. This finding may be because AOD can be disconnected from $PM_{2.5}$ in a variety of ways. For example, daytime-only measurements such as AOD would be unable to capture evening buildup

of $PM_{2.5}$ that we often saw in the Denver pilot deployment. Furthermore, $PM_{2.5}$ and AOD share a nonlinear relationship that can be altered by the aerosol hygroscopicity, aerosol vertical profile, size distribution, and chemical composition. However, AOD would likely provide greater predictive skill in the spatial variability of long-term averages in $PM_{2.5}$ (Hu et al., 2014; Liu et al., 2005; van Donkelaar et al., 2010) and locations where $PM_{2.5}$ is driven by daytime variability (van Donkelaar et al., 2011).

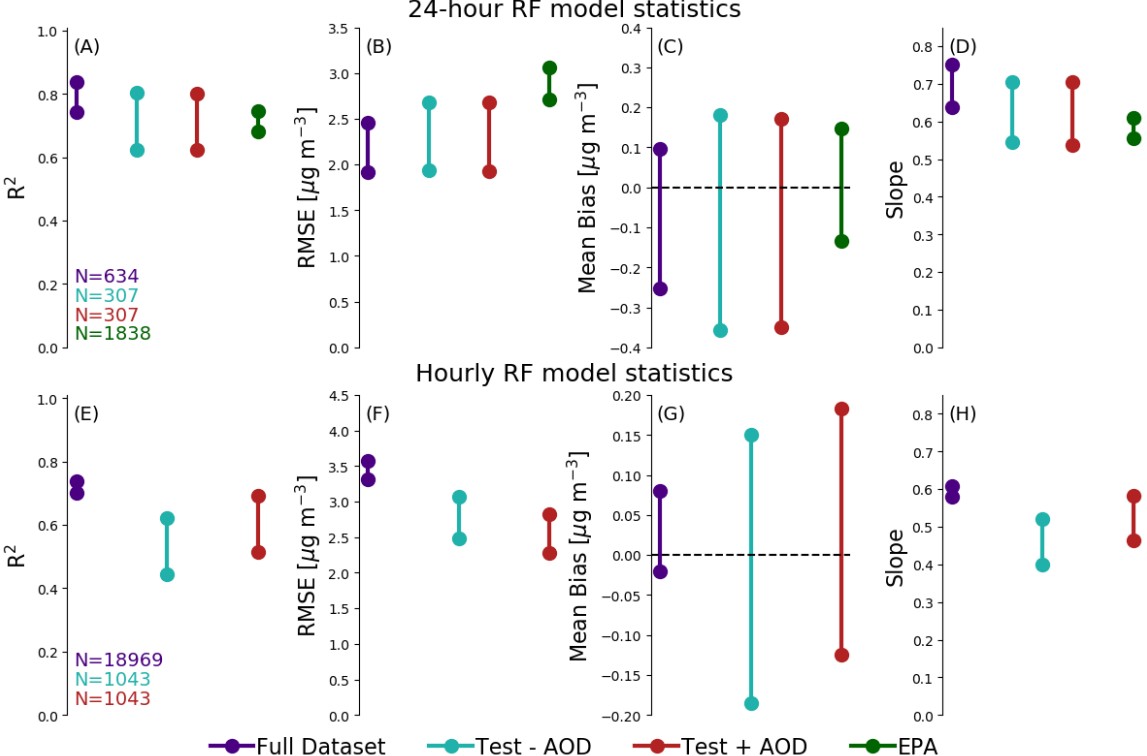

Figure 6. The 95% confidence interval of the error metrics for all of the CEAMS RF models (Full Dataset, Test - AOD, and Test + AOD) in predicting both 24-hour and hourly PM$_{2.5}$ and the error metrics for the 24-hour EPA model. The 95% confidence intervals show an estimate of the uncertainty range and, thus, if the intervals of two different models overlap, any difference in their error metrics are likely not statistically significant. The error metrics for each 24-hour PM$_{2.5}$ RF model includes (a) the coefficient of determination (R$^2$) (b) root mean squared error (RMSE), (c) mean bias, (d) and slope of the linear regression. Plots (e), (f), (g), and (h) show analogous results but for the hourly PM$_{2.5}$ predictions, which we did not predict for the EPA dataset. The size of each 24-hour and hourly dataset, before being split into *k*-folds, is shown in the bottom left corner of plot (a) and (e).

### 3.3. Variable importance for spatiotemporal PM2.5 predictions

We use our RF models not only to estimate PM$_{2.5}$ concentrations but also to investigate the variables importance in predicting PM$_{2.5}$ for wintertime in Denver. We show distributions of permutation importance for the top 10 predictors, ranked by their median permutation importance, of each RF model that predicted CEAMS and EPA PM$_{2.5}$ concentrations (Fig. 7). We found that the meteorological predictors vary largely between 24-hour (Fig. 7a-d) and hourly (Fig. 7e-g) models of PM$_{2.5}$. The daytime (11am-3pm) averaged PBLH and RH were consistently strong predictors in each CEAMS 24-hr RF model (Fig. 7a-c) and daytime PBLH was the strongest predictor in the EPA model (Fig. 7d). It was not surprising that PBLH was a strong predictor of PM$_{2.5}$, though we expected nighttime PBLH to be a stronger predictor than daytime PBLH because high PM$_{2.5}$ usually occurs during the late evening to early morning hours (Fig. S10). However, it may be that low daytime PBLH values were better correlated with periods where PM$_{2.5}$ was elevated for extended periods of time, because ventilation of air pollution was hampered by stagnant air masses. Additionally, day and night PBLH are correlated so day PBLH may act to predict





nighttime PM$_{2.5}$ buildup (Fig. 5). The strength of daytime-averaged RH in our CEAMS RF models may be due to physical
connections between PM$_{2.5}$ and RH, because high RH is tied to colder conditions (Fig. 5), which is subsequently correlated

with boundary layer heights (Fig. 5). However, this may also be due to residual bias of the Plantower measurements for which
the Barkjohn and Clements (2020) correction was unable to account. The nighttime (11pm-3am) averaged cloud fraction was
the third most important predictor in the CEAMS Full Dataset while daytime longwave net radiation was the third most
important for the CEAMS Test - AOD and Test + AOD models. As expected, since we saw no change in prediction skill
between the CEAMS Test - AOD and Test + AOD 24-hour predictions (Fig. 6a-d), AOD also did not have high permutation

importance in the 24-hour CEAMS Test + AOD model.

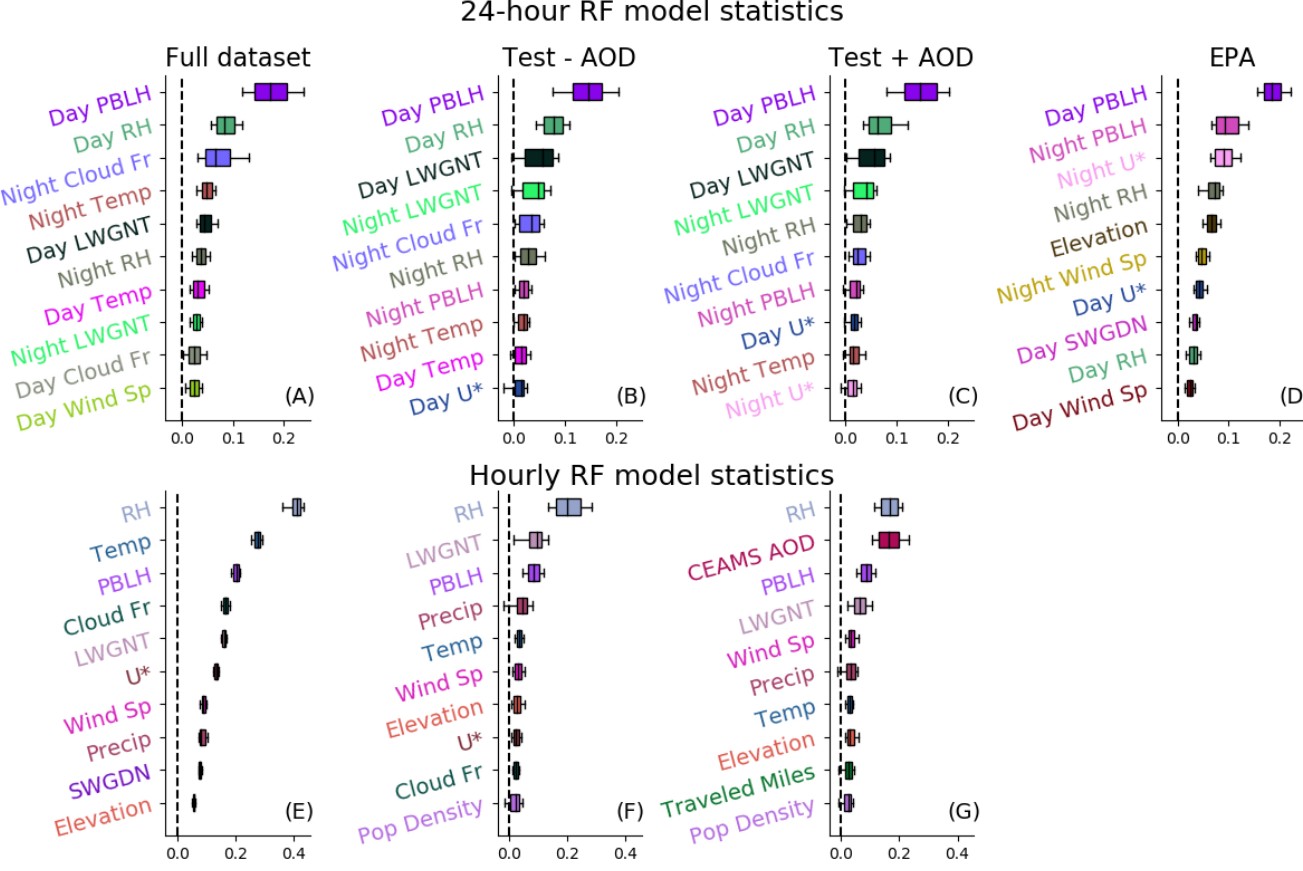

**Figure 7. Box-and-whisker plots of the distribution of 500 permutation importance measurements from the top 10 predictors of each**
**model. The 500 permutation importance values are taken from 100 repeats of permutation importance from each of the 5 testing**
**folds. The whiskers of each box are the 10th and 90th percentile of the permutation importance distribution. The edges of each box**
**represent the 25th and 75th percentile and, finally, the centerline of each box represents the median of the permutation importance**
**distribution. (a) The 24-hour PM$_{2.5}$ predictions of the CEAMS Full Dataset, (b) The 24-hour PM$_{2.5}$ predictions of the CEAMS Test**
**- AOD dataset, (c) The 24-hour PM$_{2.5}$ predictions of the CEAMS Test + AOD dataset. Plots (d), (e), and (f) are analogous to the**
**CEAMS 24-hour PM$_{2.5}$ RF models in plots (a), (b), and (c) but for hourly averaged PM$_{2.5}$ and associated predictors. The permutation**
**importance of every predictor used for each model is in the Supplement (Fig. S15-S16).**



The hourly RF models relied more on different meteorological variables than the 24-hour models (Fig. 7e-g). The most important predictor for all three hourly models was hourly averaged RH. We hypothesize that RH is the strongest predictor because the RH correction factor that we applied to our hourly-averaged $PM_{2.5}$ data is based on Barkjohn and Clements (2020),

which used 24-hour averaged $PM_{2.5}$. Thus, the importance of RH in our model may be more reflective of the RH bias in the sensor measurement than the physical connection between $PM_{2.5}$ and RH. Unlike the 24-hour CEAMS models, hourly-averaged AOD was the second strongest predictor in our Test + AOD model, and we saw improvement in mean prediction skill of this model compared to the hourly Test - AOD model (Fig. 7e-h). This result implies that hourly AOD added some skill in predicting hourly $PM_{2.5}$ data. The increased importance of AOD in the hourly models relative to the 24-hour models is

likely because the AOD is co-located in time (within the hour) with the $PM_{2.5}$ measurement but not with most of the 24-hour period (as AOD is only available during daylight hours).

Finally, we expected spatial predictors such as elevation, vehicle miles traveled, and population density to be more important for all of the RF models, because we hypothesized that air pollution would pool at low elevations during the winter in Denver,

as late evening traffic and residential wood burning emissions were trapped over Denver by stagnant air. Instead, we found that their permutation importance was near zero for all of these variables in the CEAMS models (Fig. S15-S18). However, our EPA model results indicated that elevation was a moderately important predictor when we used shuffled $k$-folds (Fig. 7d) and the 2nd most important when we used consecutive $k$-folds (Fig. S19). Vehicle miles traveled and population density are generally not important predictors in our RF models, which may be due to temporal variability being larger than spatial

variability in our dataset as well as these predictors having no correlation with $PM_{2.5}$ (Figure 5).

### 3.4. Sensitivity of Results to Data Processing and RF Setup Decisions

One critical caveat to our CEAMS data analysis is that there is strong autocorrelation in daily, and especially hourly, $PM_{2.5}$. Thus, when we shuffle the $PM_{2.5}$ data and their associated predictors before splitting the data into $k$-folds for training and testing, information will be shared between the training and testing datasets. We tested the potential impact of autocorrelation

on our model skill by repeating the CEAMS and EPA analyses without shuffling the data before splitting it into $k$-folds. We saw a significant decrease in skill for the CEAMS RF models, especially hourly (Fig. S12), and a decrease in the consistency of predictor ranking (Fig. S17-S18). Our analysis of the EPA RFs, however, showed a smaller decrease in predictive skill when we compared the results from models trained and tested using shuffled vs. consecutive $k$-folds. For example, the upper bound of the 95% confidence intervals decreased by 0.5 and 0.3 for the CEAMS Full Dataset and EPA 24-hour models,

respectively, when we used consecutive $k$-folds. Furthermore, the meteorological and geographical predictors remained more consistent in the EPA model when we used consecutive $k$-folds (Fig. S19). To test whether these results were due to the increased length of the EPA dataset, we repeated the analysis only using EPA measurements from 15 November 2019 - 15 January 2020, the same time period as our CEAMS deployment, and found similar results for both shuffled (Fig. S20) and





consecutive (Fig. S21) *k*-folds. Thus, the sharp decrease in CEAMS RF skill may be due to the quality of the Plantower sensor
measurements and/or the inconsistent sampling patterns of the CEAMS AMOD citizen science deployment, which means that
the model would not be able to train itself appropriately to compare well to unseen data. Furthermore, as we mentioned in our
discussion of Figure 2, there are only a few short periods of significantly elevated $PM_{2.5}$ during the CEAMS deployment,
which led to consistent under-prediction of high $PM_{2.5}$ in the RF models (Fig. 7d and 7h), especially in those that used
consecutive *k*-folds (Fig. S12). However, even though we do not have confidence that our CEAMS model would have
predictive skill for new time periods, we do have more confidence that our interpretation of the top meteorological and
geographical relationships is valid under the conditions of the CEAMS campaign.

In addition to the impact of autocorrelation and shuffling on our results, we found that various decisions made in processing
our data could lead to variations in the predictive skill of our models and the order of variable importance's. For example, we
found that using a linear interpolation method instead of nearest neighbor for co-locating the GEOS-FP meteorological data to
the CEAMS monitors affected which predictors were considered most important (not shown here), likely because the linear
interpolation method introduced greater spatial variability among predictors when comparing $PM_{2.5}$ from monitors in the same
grid-box. We also saw that our results were sensitive to the use of an RH correction for the CEAMS $PM_{2.5}$ because the relative
importance of RH variables decreased after the RH correction was applied in the 24-hour CEAMS RF models. Finally, we
found it useful to tune our models on a greater selection of hyperparameters than the maximum depth and the number of trees.
We recommend that future investigations of $PM_{2.5}$ with machine learning (RF in particular) carefully consider the decisions
described above.

## 4. Conclusions

The CEAMS pilot campaign provided a novel high-spatial-density, low-cost network of citizen-scientist-deployed monitors
that captured coincident sub-hourly $PM_{2.5}$ and AOD measurements in Denver. For the measurements gathered in this work in
Denver over wintertime, $PM_{2.5}$ concentrations varied much more with time than in space. This finding, that $PM_{2.5}$ varies less
with space than time within an urban environment, is generally consistent with recent $PM_{2.5}$ measurements made in other US
cities including Oakland, CA (e.g. Shah et al., 2018) and Pittsburgh, PA (Gu et al., 2018).

To understand potential drivers of $PM_{2.5}$ over wintertime Denver, we analyzed the importance of various meteorological and
geographical features in predicting spatiotemporal variability of $PM_{2.5}$ from both the CEAMS low-cost and EPA reference
networks. We found that daytime-averaged (11am-3pm) PBLH was the strongest predictor of intra-city spatiotemporal
variability for both low-cost and reference measurements of 24-hour averaged $PM_{2.5}$. The ranking of less important predictors
in our CEAMS and EPA RF models differed, however. For example, nighttime-averaged PBLH and friction velocity were
strong predictors of EPA 24-hour averaged $PM_{2.5}$, while daytime-averaged RH was a strong predictor of CEAMS 24-hour





averaged PM$_{2.5}$. We also found that hourly averaged RH was the strongest predictor of CEAMS hourly-averaged PM$_{2.5}$. However, we expect that the RFs' reliance on RH for CEAMS 24-hour and hourly PM$_{2.5}$ prediction was likely due, in part, to residual RH bias in the Plantower measurements of PM$_{2.5}$, especially since RH was not one of the top 3 predictors in our EPA RF model.


Spatial variables such as population density and number of vehicle miles traveled were consistently unimportant predictors in our RF models, although elevation was important in our multi-year EPA model. Perhaps due to the lack of importance placed on spatial variables, our RF models were unable to fully capture the extent of the spatial variability of PM$_{2.5}$ seen over Denver (Fig. S12-S13). Historically, most LUR modeling has relied on spatial variables to explain differences in PM$_{2.5}$ concentrations

and discounted temporal variability since the objective is usually to quantify the average PM$_{2.5}$ exposure over a time period of interest (e.g., seasonal, annual). In cases where studies have developed spatiotemporal LUR models because there is an interest in quantifying the time-resolved PM$_{2.5}$ exposure (Martenies et al., 2021), they do not appear to use meteorological variables directly. This work suggests that LUR modeling can benefit from using meteorological variables (e.g., PBLH) in addition to spatial and geographical variables in estimating PM$_{2.5}$.


Finally, we tested whether coincident AOD measurements added predictive skill to hourly and 24-hour averaged PM$_{2.5}$ predictions beyond what was achievable using only geographical and meteorological information in wintertime Denver, as may be possible with satellite AOD retrievals. We found that daily-averaged AOD measurements did not improve RF model predictions of CEAMS 24-hour PM$_{2.5}$, nor was AOD identified as a strong predictor of 24-hour PM$_{2.5}$ based on the permutation

metric. The lack of skill added by AOD to 24-hour PM$_{2.5}$ prediction is likely because 24-hour PM$_{2.5}$ in wintertime Denver is largely driven by evening and overnight build-up of air pollution, which daytime-only measurements such as AOD cannot capture. However, when incorporating CEAMS AOD as a predictor in our RF model of hourly-averaged PM$_{2.5}$, we found an increase in average prediction skill, and the hourly averaged AOD was the second strongest predictor based on a permutation importance metric  (although the 95% confidence intervals overlapped, which implies that the increase in model skill had low

statistical significance). This implies that AOD retrieved from geostationary satellites may be a better predictor of PM$_{2.5}$ than AOD from polar-orbiting satellites, because they may help capture more of the diurnal cycle of aerosols. We also expect AMOD AOD to be a better predictor of daily and hourly averaged PM$_{2.5}$ in other seasons or locations where enhanced PM$_{2.5}$ is not driven as strongly by nighttime conditions.

The CEAMS deployment in Denver for the winter of 2019-2020 was hampered by inconsistencies in sampling locations, sampling times, and machine errors, which resulted in a limited dataset. Despite these setbacks, this deployment provided a novel dataset that informed us about possible interactions between meteorological and geographical variables, as well as the potential for low-cost AOD measurements to aid in the prediction of high-resolution spatiotemporal variability in PM$_{2.5}$. We recommend that future work mostly concerned about predicting high PM$_{2.5}$ days in cities consider using classification RF





models that only work to predict "low" and "high" PM$_{2.5}$ days. This may provide more insight into how different spatiotemporal predictors play a role in elevated PM$_{2.5}$ events. We also recommend that future work incorporate a more thorough list of spatial predictors or use a hybrid approach that combines traditional LUR techniques and ML, such as Considine et al. (2021), to improve predictions of spatial variability for sub-city PM$_{2.5}$.

**Acknowledgements**

This research was supported by the National Aeronautics and Space Administration (NASA) grant number 80NSSC18M0120 and NASA Health and Air Quality Applied Sciences Team grant number 80NSSC21K0429. We are grateful for the CEAMS citizen scientists who participated in gathering the data used in this study as well as to Katelyn O'Dell, John Mehaffy, and the CSU Stats Helpdesk for their assistance in preparing data for Random Forest model analysis. The GEOS-FP data used in this study/project have been provided by the Global Modeling and Assimilation Office (GMAO) at NASA Goddard Space Flight
Center.

**Code/Data Availability**

All air quality data collected and used for this study during the CEAMS Denver, CO deployment are available at the following URL: https://hdl.handle.net/10217/233884

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
