# Peer review of "Technical note: Investigating sub-city gradients of air quality: lessons learned with low-cost PM2.5 and AOD monitors and machine learning"

_Atmospheric Chemistry and Physics, 2021_

## Author Comment (AC1)

We thank the referees for their insightful comments (*italic black*), which have allowed us to produce a stronger manuscript. Our responses to the major and minor comments are given below in black. Changes to the original manuscript text are in orange and text reproduced from the first submission is in blue. We've included a Microsoft Word document that tracks all the changes that we've made in response to the reviewer comments. Many changes in the document were made to limit the manuscript to the required word count instead of in direct response to the reviewers and thus, are not listed here. Line numbers in the following document are associated with the Word document.

**Reviewer 1:** This paper uses the machine learning (ML) method to investigate the spatiotemporal variability of $PM_{2.5}$ in winter over Denver. Although this is an interesting attempt, I found the methodology of this study has not been clearly stated, so that I can't confirm the results are scientifically sound under current conditions.

Reviewer 1's Comments:
1. *Comment: Section 2.1.3. Question on the spatial resolution of the meteorological inputs. The meteorological inputs for the RF model are derived from the GESO-FP data with a coarse resolution of 25 km. According to fig.1, almost 2/3 of the sites are located in one grid. You mention that the data were interpolated spatially to the CEAMS and EPA sites, but no detailed information is given. What method do you use to downscale the areal data into point data? How do you check the accuracy of the interpolation results?*

Response: Yes, the GEOS-FP data have a resolution of 25 km and many of our sites were in one grid. We spatially interpolated the meteorological inputs using a linear interpolation function (SciPy griddata). We also tested our model using a nearest-neighbor interpolation of the inputs, which showed similar conclusions. The major difference using nearest-neighbor interpolation instead of linear, was that nighttime cloud fraction was more important in the CEAMS 24-hour Full Dataset RF model and CEAMS AOD was less important in the CEAMS Test + AOD hourly RF model. Those results are now shown in the supplement (Figure S23-S24). We updated the text in Section 2.1.3: "Finally, all of the GEOS-FP variables were linearly interpolated spatially from the gridpoint centers to the CEAMS and EPA monitor locations using the SciPy griddata function (Virtanen et al., 2020) to better relate $PM_{2.5}$ observations with the environment for RF predictions. To show the impact of different spatial interpolation techniques, we repeated our analysis using nearest-neighbor interpolation, which resulted in similar conclusions in the Supplement." We also updated the text in Section 3.4: "For example, we found that using a linear interpolation method instead of nearest-neighbor for co-locating the GEOS-FP meteorological data to the CEAMS monitors resulted in minor changes to the order of predictors, such as a decreased importance of nighttime cloud fraction for the CEAMS 24-hour Full Dataset RF model (Figure S23-S24). These changes are likely because the linear interpolation method introduced greater spatial variability among predictors when comparing $PM_{2.5}$ from monitors in the same grid-box."

2. *Comment: Section 2.2.3 and figure 6. Question on the validation process of the ML model. As far as I understand, you use the whole dataset to tune the RF model with k-fold cross validation method, and then use the same dataset to validate the model performance with*

*k-fold cross validation method and bootstrapping. In my opinion, to give unbiased evaluation on the robustness of the ML model, the validation dataset should never be used in the training process. Otherwise, the accuracy of the ML model is certain to be high since the model has already learned the pattern. Please clarify if my understanding is wrong.*

*Line 464-466: "However, even though we do not have confidence that our CEAMS model would have predictive skill for new time periods, we do have more confidence that our interpretation of the top meteorological and geographical relationships is valid under the conditions of the CEAMS campaign."*

*I do not agree with this sentence. A well-developed ML model should be able to work on new datasets. This is why we test the model's ability with new datasets in the model validation process. If the model can only work well on the training dataset, it may have an overfitting problem.*

Response: The reviewer makes an excellent point here. Due to our limited dataset and the unique goals of this project (determining the main drivers/predictors of air pollution events in Denver as opposed to building the best model for prediction), we did use the whole dataset for tuning, training and validation. However, we have taken the reviewer's suggestion and changed our methods. We use the first 30% of our dataset just for model tuning, and then we use the other 70% of the dataset for training and cross validation testing. This processing method had little impact on the conclusions of the paper, though it did change the order of some of the predictors. To highlight this, we removed the following line from the original text: "Typically, to ensure against over or underfitting, the data are split into separate tuning, training, and testing datasets. However, since our dataset spans only a couple months, we tuned, trained, and tested our RF models using a cross validation (CV) method." We then replaced it with this line: "We tuned our model to determine the optimal hyperparameters using a cross validation method on the first 30% of each dataset. The following 70% of each dataset was used to train and test the models using a separate (CV) method."

We have also updated the results (Fig. 6-7) in the manuscript to reflect the changes in the data processing and several figures in the Supplement. Changing our data processing methods had some impact on the relative skills of each of our models resulting in a change to Figure 6 in the main text:

Original Figure 6 from the manuscript:

[Figure]

Updated Figure 6:

[Figure]

The changes to our model skill in Figure 6 resulted in the following text changes in Section 3.2: "The 95% confidence intervals of the $R^2$ and slope metrics from the Full Dataset model did not overlap with the Test - AOD and Test + AOD RF models, which implies that limiting the CEAMS data to locations and days where AOD was available did result in a small reduction in model skill for 24-hour $PM_{2.5}$ prediction. We also saw a small reduction in skill for hourly $PM_{2.5}$ predictions (Fig. 6e-g) after limiting the dataset to only locations and hours where AOD was also taken. The 24-hour CEAMS models showed similar skill to the EPA model for most metrics (Fig. 6a-d). However, results for the RF models using consecutive $k$-folds showed a significant decrease in prediction skill while the EPA model results showed a less substantial decrease in skill (Fig. S12)." There were other small changes to the text about model skill that can be seen clearly in the tracked changes document.

Our changes in the tuning-training methodology resulted in small changes to the order of some our predictors. Thus, we have also updated Figure 7 as follows:

Original Figure 7 from the manuscript:

[Figure]

Updated Figure 7:

[Figure]

To account for the changes in the order of the predictors, we have updated the text in Section 3.3: "The nighttime (11pm-3am) averaged cloud fraction was the third most important predictor in the CEAMS Full Dataset while nighttime PBLH was the third most important for the CEAMS Test - AOD and Test + AOD models. As expected, since we saw no change in prediction skill between the CEAMS Test - AOD and Test + AOD 24-hour predictions (Fig. 6a-d), AOD also did not have high permutation importance in the 24-hour CEAMS Test + AOD model.

The hourly RF models relied more on different meteorological variables than the 24-hour models such as hourly-averaged temperature and longwave radiation (Fig. 7e-g). However, RH was in the top three predictors for all three hourly models and PBLH was important for the Test - AOD and Test + AOD models. We hypothesize that RH is a strong predictor because the RH correction factor that we applied to our hourly-averaged $PM_{2.5}$ data is based on Barkjohn and Clements (2020), which used 24-hour averaged $PM_{2.5}$." There are similar changes in the text throughout in our discussion of predictor variables that can be seen clearly in the tracked-changes document.

3.  *Comment: Question on the temporal resolution of the inputs and outputs. In the 24-hour RF model case, the model inputs and outputs are not of the same temporal resolution. The output/prediction is 24-hour $PM_{2.5}$. But the meteorology inputs, separated into daytime (11am-3pm) group and nighttime (11pm-3am) group, only cover the information of these 8 hours of a day. This method is valid if you can prove that the 8-hour data is enough to represent the whole day. Since these comments are related to the fundamental methodology*

*of the study, I cannot recommend this study for publication before these questions are explained.*

Response: We agree that the reasoning was not clearly stated in the manuscript for using these variables with different temporal resolutions. $PM_{2.5}$ is regulated as a 24-hour average, which is why our model predicts a 24-hour average. However, $PM_{2.5}$ emissions do not have a constant emission rate, and instead have substantial diurnal variability which can interact with the meteorological variables that also have diurnal variability. For examples see the figure below, where we plot the average diurnal cycles of 3 GEOS-FP variables co-located to the site of one of our AMODs:

[Figure]

Thus, our goal was not to show that an 8-hour average of data represents the whole day, but to determine if the meteorological variables during certain periods of the day are better predictors of air quality than a daily average. Many of the meteorological variables have strong diurnal cycles in wintertime (for example, temperature, boundary layer height) which may impact average $PM_{2.5}$ concentrations. Using the two 4-hour averages allows us to investigate separate daytime and nighttime conditions instead of smoothing out the predictors with a 24-hour mean. For example, we are interested in understanding if the height of the boundary layer during the night (i.e., trapping pollution) is a better predictor of the PM2.5 concentrations than the height of the boundary layer during the day (i.e., mixing out of pollution). We have added the following sentences to Section 2.2.1 to provide better motivation for our temporal resolution: "In this study, we created RF models

with the scikit-learn Python package (Pedregosa et al. 2011) to predict the spatial and temporal variability of $PM_{2.5}$ using the predictors in Table 1. As shown in Table 1, we used separate daytime (11am-3pm) and nighttime (11pm-3am) averages of many of our meteorological predictors. Many meteorological variables have strong diurnal cycles in wintertime (e.g., temperature, boundary layer height), which may impact average $PM_{2.5}$. Thus, using two 4-hour averages allows us to investigate the separate daytime and nighttime conditions, and their relation to 24-hour $PM_{2.5}$, instead of smoothing out the predictors using a 24-hour mean."

4. *The target of this study is to "investigate the potential drivers of fine-scale $PM_{2.5}$ spatiotemporal variability in wintertime Denver…" (line 93). However, you pay a lot of attention on testing the importance of including co-located AOD measurements in the RF model. What is the reason of picking this specific variable out of all factors that could contribute to the spatiotemporal variation of $PM_{2.5}$? The motivation sounds weak especially when your conclusion is that adding co-located AOD data makes very little improvements to model prediction (line 508-509 and line 514-515).*

Response: Aerosol Optical Depth has been used to predict surface $PM_{2.5}$ concentrations extensively in the literature, both for air pollution and health studies (including the Global Burden of Disease). The device used in this study (AMOD) was specifically developed to collect more colocated $PM_{2.5}$ and AOD measurements that could be used to analyze the types of questions we pose here about the suitability of using AOD to predict $PM_{2.5}$ concentrations at finer scales. We have added the following lines to the introduction to further emphasize the importance of testing AOD as a predictor for $PM_{2.5}$: "AOD data is commonly used to aid in global $PM_{2.5}$ predictions (i.e., Hammer et al., 2020; Liu et al., 2004, 2005; van Donkelaar et al., 2006, 2010, 2013), however, fewer studies have assessed how $PM_{2.5}$ and AOD are related at finer spatial and temporal resolution. In this study, we test whether co-located AOD measurements are identified as an important predictor of $PM_{2.5}$ and whether they increase the overall RF prediction skill compared to RFs that only used geographic and meteorological variables."

5. *2) Line 149-150: "In this study, the Plantower $PM_{2.5}$ data were not corrected using the time-integrated filter measurements of $PM_{2.5}$ taken by the AMODs as in Ford et al., (2019)". Did you compare the real-time measurement with time-integrated filter measurements? Are they in good agreement? The word "corrected" sounds that the real-time measurement is not so reliable as the filter measurements. Please rewrite it.*

Response: In our previous study, we found good agreement between the filter and Plantower measurements when the filter mass was above the detection limit (as discussed in Ford et al., 2019). However, for this wintertime deployment in Denver, the $PM_{2.5}$ mass on the filter was often below the limit of detection. Additionally, many of the filters had been compromised (mishandled by participants) before the mass could be estimated. Thus, we did not use them to correct the low-cost $PM_{2.5}$ sensor. We have removed this line in the manuscript, as it is not relevant to our analysis. However, we do discuss our method of correcting the Plantower measurements in Section 2.1.1, which relies on the correction method in Barkjohn et al., 2021, which has been used widely in the literature (e.g., O'Dell et al., 2022).

6.  *Line 372-373: "We also found that the RF models were better at capturing temporal variability than spatial variability during the CEAMS deployment." Is Figure S13 the average results of all available monitoring sites? If so, I can only see the model's ability on temporal variability but not on spatial variability. Please give more explanation on this finding.*

Response: We agree that we did not clarify Figure S13 enough in the main text. Figure S13 is showing the standard deviation across all sites for each hour that had at least 10 monitors active, as opposed to each individual site. Thus, this figure shows that predicted spatial variability of $PM_{2.5}$ across sites was consistently lower than measured variability, especially when $PM_{2.5}$ was elevated. We have updated the text in Section 3.2: "The hourly $PM_{2.5}$ observations showed an average spatial standard deviation of ~2.8 $\mu$g m$^{-3}$ when we found the standard deviation across all sites for hours with at least 10 monitors simultaneously taking measurements. Contrastingly, RF model predictions showed an average spatial standard deviation of ~1.7 $\mu$g m$^{-3}$ for shuffled $k$-folds (Fig. S13) and only ~0.7 for consecutive $k$-folds (Fig. S14)."

**Reviewer 2:** The authors propose a Random Forest model to predict sub-city-scale PM$_{2.5}$ concentrations. The studied case is wintertime in Denver, captured by CEAMS low-cost sensor network on the one hand, and EPA's reference monitors on the other. A permutation metric is applied to conclude predictor importance, with a special interest in AOD. While this is an interesting approach to quantify the influence of various drivers, I would like to point out some insufficiently discussed choices in applying the methods that might compromise the results.

Reviewer 2's Comments:

1. *Comment: From line 283 I conclude that the model was trained and tested on the same dataset that was used to tune the hyperparameters beforehand. Therefore, the test data can't strictly be considered unseen. The extent to which this limits the detection of overfitting and therefore validity of the results should at least be discussed. Potential overfitting is also implied by the authors' lack of confidence in the predictive skills of their model for new data (lines 464-466).*

Response: Reviewer 1 made the same comment and both are greatly appreciated. Due to our limited dataset and the unique goals of this project (determining the main drivers/predictors of air pollution events in Denver as opposed to building the best model for prediction), we did use the whole dataset for tuning, training and validation. However, we have taken the reviewers' suggestions and changed our methods. We use the first 30% of our dataset just for model tuning, and then we use the other 70% of the dataset for training and cross validation testing. This change in methodology had little impact on the conclusions of the paper, though it did change the order of some of the predictors. To highlight this, we removed the following line from the original text: "Typically, to ensure against over or underfitting, the data are split into separate tuning, training, and testing datasets. However, since our dataset spans only a couple months, we tuned, trained, and tested our RF models using a cross validation (CV) method." We then replaced it with this line: "We tuned our model to determine the optimal hyperparameters using a cross validation method on the first 30% of each dataset. The following 70% of each dataset was used to train and test the models using a separate (CV) method." We have updated all the results figures (Figures 6-7), several supplementary figures, and made changes to the text to reflect the changes in the data processing. For example:

Original Figure 6 from the manuscript:

[Figure]

Updated Figure 6:

[Figure]

The changes to our model skill in Figure 6 resulted in the following text changes in Section 3.2: "The 95% confidence intervals of the $R^2$ and slope metrics from the Full Dataset model did not overlap with the Test - AOD and Test + AOD RF models, which implies that limiting the CEAMS data to locations and days where AOD was available did result in a small reduction in model skill for 24-hour PM$_{2.5}$ prediction. We also saw a small reduction in skill for hourly PM$_{2.5}$ predictions (Fig. 6e-g) after limiting the dataset to only locations and hours where AOD was also taken. The 24-hour CEAMS models showed similar skill to the EPA model for most metrics (Fig. 6a-d). However, results for the RF models using consecutive *k*-folds showed a significant decrease in prediction skill while the EPA model results showed a less substantial decrease in skill (Fig. S12)." There were other small changes to the text about model skill that can be seen clearly in the tracked changes document.

The changes in our tuning-training methodology also had small impacts on the order of predictors seen in Figure 7 of the main text. Thus, see below the original Figure 7 from the manuscript:

[Figure]

Updated Figure 7:

[Figure]

To account for the changes in the order of the predictors, we have updated the text in Section 3.3: "The nighttime (11pm-3am) averaged cloud fraction was the third most important predictor in the CEAMS Full Dataset while nighttime PBLH was the third most important for the CEAMS Test - AOD and Test + AOD models. As expected, since we saw no change in prediction skill between the CEAMS Test - AOD and Test + AOD 24-hour predictions (Fig. 6a-d), AOD also did not have high permutation importance in the 24-hour CEAMS Test + AOD model.

The hourly RF models relied more on different meteorological variables than the 24-hour models such as hourly-averaged temperature and longwave radiation (Fig. 7e-g). However, RH was in the top three predictors for all three hourly models and PBLH was important for the Test - AOD and Test + AOD models. We hypothesize that RH is a strong predictor because the RH correction factor that we applied to our hourly-averaged PM$_{2.5}$ data is based on Barkjohn and Clements (2020), which used 24-hour averaged PM$_{2.5}$." There are similar changes in the text throughout in our discussion of predictor variables that can be seen clearly in the tracked-changes document.

2. *Caveats in the analysis of predictor importance. A citation introducing and discussing the permutation metric seems to be missing. To my knowledge, the current gold standard to deduce predictor importance are Shapley-value based methods, due to their favorable theoretical properties. Therefore, it would be nice to justify the choice (presumably computational cost?). Especially the presence of a competitor like RH, that apparently got an unfair advantage by the correction factor (lines 428-430), seems to call for a metric where subsets of predictors are left out in the training. It is also questionable how well*

*models trained on highly autocorrelated data are suited for the importance analysis, as stated in lines 314-316. Further justification is needed.*

Response: We thank the reviewer for this comment. To be clear, the correction factor was added to reduce to the bias from the low-cost CEAMS PM monitor, which has been shown to have an RH bias if uncorrected. We have added a reference for permutation importance into the following line: "Finally, to investigate the relative importance of each predictor for the RF predictions, a permutation importance metric was used, which tests the change in model prediction skill after randomly shuffling one predictor of the validation data at a time (Breiman et al., 2001)."

We also repeated the analysis with Shapely values and found similar results. We indicate this by adding the following lines: "To test the robustness of each permutation importance score, the metric was calculated 100 times for each predictor for each of the 5 iterations of the 5-fold CV, resulting in a distribution of 500 permutation importance scores per predictor. Furthermore, we estimated predictor importance using Shapley values, a method based on cooperative game theory (Lundberg and Lee, 2017), which were in general agreement with the permutation metric and shown in the Supplement."

We also added the following lines to the results in Section 3.3: Vehicle miles traveled and population density are generally not important predictors in our RF models, according to the permutation importance metric, which may be due to temporal variability being larger than spatial variability in our dataset as well as these predictors having no correlation with $PM_{2.5}$ (Figure 5). In contrast, Shapley values indicate that elevation was the most important feature in the EPA shuffled $k$-fold model, while vehicle miles traveled and population density were moderately important (Figure S.20). This difference is likely because Shapley estimates are based on the model attribution of various predictors instead of change in total skill. Thus, the magnitude of elevation had a relatively large impact on the predicted magnitude of the $PM_{2.5}$, even if it didn't have a large impact on model skill. Other than the increased importance of elevation for the EPA model, the Shapley values largely agreed with the top predictors found using the permutation metric for all models.

3. *Comment: Further investigation of the impact of interpolating the data could be insightful.*

Response: Reviewer #1 had a similar comment which we address above by adding results to the supplement using nearest neighbor interpolation instead of Linear interpolation (Figure S23-S24). Using nearest-neighbor interpolation resulted in similar conclusions as linear interpolation, except nighttime cloud fractions was considered more important in the 24-hour CEAMS Full Dataset RF model and CEAMS AOD was considered less important in the hourly Test + AOD model.

4. *Comment: To me, the main purpose of the paper is partly unclear. While transparency about the training and tuning process is important, the extensive explanation of Random Forests, cross validation and parameter tuning seems a bit convoluted for a paper whose foremost goal is to investigate the impact of different factors on the spatiotemporal variability of PM5, and not necessarily to serve as a guide on applying RF models.*

Response: We appreciate the reviewers comment here and we agree that the objectives of the paper were not clear enough. We wrote this manuscript with the intention that it could also serve as a guide to aid others using Random Forest models to study air pollution. We added the following lines to the end of the introduction to clarify the primary objectives of this paper: "Finally, we discuss our RF methods in detail and discuss how decisions made during data processing and model configuration may have influenced our results and the subsequent interpretation. Thus, our objectives in this study are threefold: 1) investigate meteorological and geographic drivers of sub-city $PM_{2.5}$; 2) quantify any skill added by using AOD as an additional predictor; and, 3) provide a guide for using random forests to investigate air quality measurements and share lessons we've learned."

5. *Comment: Line 160: consistency in use of special characters in "Angstrom."*

Response: We corrected this so that all examples use special character Angström

6. *Comment: Line 262: missing hyphen in "over- or underfitting"*

Response:  This sentence was removed.

7. *Comment: Line 278: "depth of 15, 2 samples needed" – as far as I know, starting a clause with a symbol is considered bad style and also interrupts the reading flow here*

Response: Thank you for pointing this out. The line now reads as: "Similarly, we chose a maximum depth of 15 for each tree, required 2 samples to form a leaf, and required 5 samples to split a branch node"

8. *Comment: Table 2: The explanation for min_samples_leaf seems misleading, since leaf nodes aren't split. Do you mean the minimum samples stored in a leaf?*

Response: Thank you, this is a good point. It has been re-written as follows: "Minimum samples needed to store in a leaf node"

9. *Comment: Line 289: "This process was repeated until a distribution of each error statistic was created" makes it sounds as if there was an absolute threshold on how often to repeat a process before you can apply statistics. Maybe rather something like: "...repeated to create a distribution..."?*

Response: The line now reads as "This process was repeated to create distributions of each error statistic"

*10. Comment: It seems counterintuitive that the shuffled folds entail more autocorrelation than the consecutive ones. A very brief explanation or some numbers in the supplementary material could be helpful. On a positive note, I appreciate the topic is addressed at all.*

Response: When the data are shuffled before being chunked into *k-folds*, each fold has some information about the entire time series because the meteorological values and $PM_{2.5}$ data are correlated in time. As a result, validation scores are boosted due to autocorrelation between the validation fold and the training folds. Thus, we present results using shuffled and consecutive *k*-folds to illustrate the difference. However, we have confidence in the predictors we identified as important despite the autocorrelation, since our analysis of the EPA $PM_{2.5}$ monitors largely agreed with the top predictors in our other models (even when we used consecutive *k*-folds). We have added to the discussion of the *k*-folds in Section 2.2.4: "Alternatively, data can be chunked into consecutive *k*-folds, which reduces information sharing between testing and training datasets. To illustrate this, we show the CEAMS $PM_{2.5}$ time-series over Denver separated into 5 folds with shuffling (Fig 3a) and without shuffling (Fig. 3b). We see in Figure 3 that each shuffled fold contains some information about the entire time series, unlike the consecutive folds."